# Identification of a Dexamethasone Mediated Radioprotection Mechanism Reveals New Therapeutic Vulnerabilities in Glioblastoma

**DOI:** 10.3390/cancers13020361

**Published:** 2021-01-19

**Authors:** Paula Aldaz, Jaione Auzmendi-Iriarte, Maika Durántez, Irene Lasheras-Otero, Estefania Carrasco-Garcia, M. Victoria Zelaya, Laura Bragado, Ana Olías-Arjona, Larraitz Egaña, Nicolás Samprón, Idoia Morilla, Marta Redondo-Muñoz, Mikel Rico, Massimo Squatrito, Marta Maria-Alonso, Joaquín Fernández-Irigoyen, Enrique Santamaria, Iñaki M. Larráyoz, Claudia Wellbrock, Ander Matheu, Imanol Arozarena

**Affiliations:** 1Cancer Signalling Unit, Navarrabiomed, Complejo Hospitalario de Navarra (CHN), Universidad Pública de Navarra (UPNA), 31008 Pamplona, Spain; paula.aldaz.donamaria@navarra.es (P.A.); maidurdel@gmail.com (M.D.); irene.lasheras.otero@navarra.es (I.L.-O.); ana.olias.arjona@navarra.es (A.O.-A.); idoia.morilla.ruiz@navarra.es (I.M.); marta.redondo.munoz@navarra.es (M.R.-M.); claudiawellbrock97@gmail.com (C.W.); 2Health Research Institute of Navarre (IdiSNA), 31008 Pamplona, Spain; mv.zelaya.huerta@navarra.es (M.V.Z.); laura.bragado.alvarez@navarra.es (L.B.); mikel.rico.oses@navarra.es (M.R.); mmalonso@unav.es (M.M.-A.); joaquin.fernandez.irigoyen@navarra.es (J.F.-I.); enrique.santamaria.martinez@navarra.es (E.S.); 3Cellular Oncology Group, Biodonostia Health Research Institute, 20014 San Sebastian, Spain; Jaione.auzmendi@biodonostia.org (J.A.-I.); estefania.carrasco@biodonostia.org (E.C.-G.); larraitz.eganaotano@osakidetza.eus (L.E.); nicolas.sampron@me.com (N.S.); ander.matheu@biodonostia.org (A.M.); 4CIBER de Fragilidad y Envejecimiento Saludable (CIBERfes), 28029 Madrid, Spain; 5Pathological Anatomy Department, CHN, IdiSNA, 31008 Pamplona, Spain; 6Department of Radiophysics and Radiological Protection, CHN, 31008 Pamplona, Spain; 7Biodonostia University Hospital, 20014 San Sebastian, Spain; 8Department of Oncology, CHN, IdiSNA, 31008 Pamplona, Spain; 9Radiotherapy Oncology Department, CHN, 31008 Pamplona, Spain; 10Seve-Ballesteros Foundation Brain Tumor Group, Cancer Cell Biology Program, Spanish National Cancer Research Centre (CNIO), 28029 Madrid, Spain; msquatrito@cnio.es; 11Program in Solid Tumours and Biomarkers, Foundation for Applied Medical Research, 31008 Pamplona, Spain; 12Department of Pediatrics, University Hospital of Navarre, 31008 Pamplona, Spain; 13Clinical Neuroproteomics Unit, Navarrabiomed, CHN, UPNA, IdiSNA, 31008 Pamplona, Spain; 14Proteored-ISCIII, Proteomics Platform, Navarrabiomed, CHN, UPNA, IdiSNA, 31008 Pamplona, Spain; 15Center for Biomedical Research of La Rioja (CIBIR), Neurodegeneration Area, Biomarkers and Molecular Signaling Group, 26006 Logroño, Spain; ilarrayoz@riojasalud.es; 16IKERBASQUE, Basque Foundation for Science, 48013 Bilbao, Spain

**Keywords:** glioblastoma, dexamethasone, PDGFR, mitosis checkpoint, sunitinib

## Abstract

**Simple Summary:**

The standard of care for patients with newly diagnosed glioblastoma (GBM) comprises surgery followed by radio- and chemotherapy. In addition, dexamethasone is used to manage the development of inflammation within the brain in general, and particularly during treatment. The effects of dexamethasone on patient survival however remain controversial because several clinical studies suggest that dexamethasone could potentially restrict effective radiotherapy. With the idea to improve GBM therapy, we set out to identify small molecule inhibitors that could improve the killing of GBM cells when applied together with radiotherapy. We have identified a novel dexamethasone-induced mechanism that can directly protect GBM cells from radiotherapy and thus may contribute to the adverse effects observed in the clinic. Strikingly, this mechanism also sensitises GBM cells to tyrosine kinase inhibitors, thus encouraging the revision of the use of these inhibitors for the treatment of GBM, potentially in an adjuvant setting.

**Abstract:**

(1) Background: Despite the indisputable effectiveness of dexamethasone (DEXA) to reduce inflammation in glioblastoma (GBM) patients, its influence on tumour progression and radiotherapy response remains controversial. (2) Methods: We analysed patient data and used expression and cell biological analyses to assess effects of DEXA on GBM cells. We tested the efficacy of tyrosine kinase inhibitors in vitro and in vivo. (3) Results: We confirm in our patient cohort that administration of DEXA correlates with worse overall survival and shorter time to relapse. In GBM cells and glioma stem-like cells (GSCs) DEXA down-regulates genes controlling G2/M and mitotic-spindle checkpoints, and it enables cells to override the spindle assembly checkpoint (SAC). Concurrently, DEXA up-regulates Platelet Derived Growth Factor Receptor (PDGFR) signalling, which stimulates expression of anti-apoptotic regulators BCL2L1 and MCL1, required for survival during extended mitosis. Importantly, the protective potential of DEXA is dependent on intact tyrosine kinase signalling and ponatinib, sunitinib and dasatinib, all effectively overcome the radio-protective and pro-proliferative activity of DEXA. Moreover, we discovered that DEXA-induced signalling creates a therapeutic vulnerability for sunitinib in GSCs and GBM cells in vitro and in vivo. (4) Conclusions: Our results reveal a novel DEXA-induced mechanism in GBM cells and provide a rationale for revisiting the use of tyrosine kinase inhibitors for the treatment of GBM.

## 1. Introduction

Glioblastoma (GBM) remains among the cancers with poorest prognosis with a median overall survival of only 15 months after diagnosis [1]. Recent efforts to understand the genetics of GBM have improved our knowledge of the molecular events leading to gliomagenesis and mutations; amplifications or deletions of genes such as *IDH1*, *NF1*, *PTEN*, *P53*, *RB1*, *PDGFRA* or *EGFR* have been identified [2,3]. Genome-wide analyses of large patient cohorts have revealed clinically relevant GBM subtypes such as classical, proneural or mesenchymal, which correlate with particular tumour microenvironments and have prognostic implications [3,4,5].

Radiotherapy is the standard of care for GBM after surgical resection, but the vast majority of patients relapse due to intrinsic or acquired resistance. Acquired resistance to radiotherapy is thought to rely on the deregulation of DNA repair mechanisms, cell cycle progression and survival pathways in GBM cells, but also on signals from the stroma, including a hypoxic extracellular environment [6]. In addition, sub-populations of undifferentiated glioblastoma stem-like cells (GSCs) that show increased resistance to radiotherapy are thought to cause tumour relapse [6].

Almost all patients with brain tumours receive corticosteroids at some point in the course of their disease [7]. Corticosteroids help control increased intra-cranial pressure based on peritumoral vasogenic edema, which contributes significantly to morbidity and occurs in >60% of GBM patients; the incidence of edema is further increased by brain surgery, radiotherapy and adjuvant chemotherapy [7]. The gluco-corticoid dexamethasone (DEXA) is the most commonly used corticosteroid for Central Nervous System-affected cancer patients with edema-associated neurological manifestations, and over 70% of patients receive DEXA while undergoing multimodal radio/chemotherapy [7]. DEXA targets macrophages and lymphocytes thus blocking the production of pro-inflammatory cytokines, modulating innate and adaptive immunity and reducing inflammation. However, several clinical studies suggest that DEXA could potentially restrict effective radio- as well as chemotherapy as they have made the observation that low steroid use during radio/chemotherapy correlated with better survival [8,9,10,11,12,13]. While clinicians consider acting on these challenges, there are currently no real alternatives for the management of intracranial hypertension or brain edema in GBM patients.

Despite improvements being made with standard of care therapies, the prognosis of patients with GBM remains poor. Molecular targeting important players in GBM could be an alternative to tackle this disease, and receptor tyrosine kinases (RTKs) such as EGFR and Platelet Derived Growth Factor Receptor Alpha (PDGFRA) have been considered as targets in trials using small molecule inhibitors, because apart from harbouring mutations, the corresponding genes are frequently amplified [2,3]. Support for the relevance of these RTKs for GBM comes from mice genetically engineered to experience deregulated PDGFR or EGFR signalling in an adequate genetic background, as this promotes gliomagenesis [14,15,16].

Despite these encouraging clinical and pre-clinical data, so far there is no breakthrough coming from RTK targeting trials, probably because, apart from restrictions for some inhibitors to crossing the blood-brain-barrier, there have been limitations through small sample size as well as great heterogeneity in disease and prior therapy. 

With the idea to improve GBM therapy, we set out to identify small molecule inhibitors that could improve the killing of GBM cells when applied concomitant to radiotherapy with the possibility that they can also function as single agent in post-radiation maintenance. We discovered that DEXA could directly act as radio-protective factor by up-regulating a PDGFR signalling cascade in GBM cells. Importantly, this DEXA induced signalling-switch produces a general vulnerability not only in GBM cells but also in GSCs towards Food and Drug Administration (FDA)-approved tyrosine kinase inhibitors (TKIs) such as sunitinib. 

## 2. Results

### 2.1. Dexamethasone Protects from Radiotherapy and Reduces Survival in GBM Patients 

To identify mechanisms of radio-protection in GBM we analysed the effect of FDA-approved drugs on the survival of T98G cells after a single high dose of radiation (Figure 1A). Intriguingly, this identified 13 members of the family of glucocorticoids (Figure 1B). Amongst the identified glucocorticoids was dexamethasone (DEXA), which is of major clinical relevance for GBM patients [7]. We therefore assessed the effect of DEXA on our patient cohort of 285 stage IV glioma patients registered at the Donostia University Hospital in San Sebastian, Spain (for details see Appendix A). DEXA administration correlated with significantly shorter overall survival after surgery, and this effect was also seen in the cohort of patients who had received radiotherapy after surgery (Figure 1C,D). Moreover, the time to relapse after radiotherapy was significantly shorter in DEXA-treated patients (Figure 1E). Our findings are supported by previous observations [10,12,17,18] and highlight the relevance of the controversial role of DEXA, but importantly the mechanism by which DEXA can induce radio-protection of GBM is unclear.

### 2.2. Dexamethasone Suppresses Genes Required for Accurate Mitosis Control

To reveal the effects of DEXA on GBM cell function, we performed RNAseq on T98G cells (Appendix A). Pathway analysis of significantly down-regulated transcripts revealed a profound effect of DEXA on genes controlling G2/M transition, mitosis and cytokinesis (Figure 2A). Analysis of essential regulators of the G2/M and spindle assembly checkpoints (PLK1, TTK/MPS1), mitotic spindle dynamics (KIF11) and sister chromatid separation (PTTG1/securin) confirmed their down-regulation in a panel of glioblastoma cell lines (Figure 2B). Reduced expression of these genes was independent of DEXA concentrations or treatment times (not shown) and was also observed in the human glioma stem-cell line GNS166 (Figure 2C). Thus, the down-regulation of mitosis-control genes appears to be a universal response of GBM cells to DEXA.

In the The Cancer Genome Atlas (TCGA) GBM patient cohort [4] mitosis-control genes are highly expressed in ~20% of tumours, where they display striking co-expression (Appendix A). However, the majority of tumours exhibit low expression of the respective genes (Appendix A). Pitter and co-workers, who had made similar observations [18], proposed that with most patients receiving DEXA, low expression of cell cycle-regulator genes might reflect a tumour response to the corticoid. Supporting this notion, tumours from lower grade glioma patients [19], who also frequently receive DEXA besides radiotherapy, display a similar expression pattern for the respective mitosis-control genes (Appendix A). 

It was also suggested that the down-regulation of cell cycle-regulator genes explains DEXA’s negative effect on patient survival, because it would result in reduced proliferation thus allowing glioma cells to escape radiotherapy-induced toxicity [18]. Indeed, low expression of a “DEXA mitosis down” signature is significantly linked to poor overall survival and faster relapse (Appendix A). Intriguingly however, exactly the opposite is observed in lower grade gliomas (Appendix A). Notably, these survival data are independent of the mutation status of *IDH1*, which is mutated in 78% of patients in the lower grade glioma cohort and 6% of the GBM cohort [4,19]. Overall, this suggests that the situation is more complex and that the outcome of reduced mitosis-control gene expression is distinct in GBM cells, where this appears to support GBM progression. 

### 2.3. Dexamethasone Drives Proliferation by Overriding Cell Cycle Checkpoints

Concomitant with the reduced expression of G2/M checkpoint genes, DEXA significantly increased the mitotic index in a panel of GBM cell lines (Figure 2D and Appendix A). This was however not linked to a mitotic arrest because DEXA treated GBM cells continued dividing (Figure 2E and Appendix A). Nevertheless, we found that cells driven into mitosis by DEXA displayed an increased amount of mitotic errors ranging from mono- or tripolar spindles, to unaligned or lagging chromosomes and chromosome bridges (Figure 2F,G, Appendix A).

Under controlled conditions such mitotic errors will activate the spindle assembly checkpoint (SAC) resulting in an extended mitotic arrest, and if the error cannot be resolved, the default response is death in mitosis [20]. This might occur in non-transformed MCF-10A cells, where DEXA, despite increasing the mitotic index, severely reduces the number of dividing cells (Appendix A). However, cancer cells frequently override the SAC, allowing them to divide with aneuploidy or return to interphase without completing cell division, thus leading to polyploidy [21,22]. The fact that DEXA-treated GBM cells continued to divide despite an increase in mitotic errors suggested that the SAC was compromised, and that DEXA may have the potential to override the SAC.

To test this idea, we treated cells with low concentrations of the microtubule-destabilising agent vincristine, which led to a profound increase in cells in mitosis within 24 h and a reduction in viable cells (Figure 2H,I, Appendix A). As positive control we used reversine, an MPS1 (TTK) inhibitor [23] that induces SAC override and polyploidy [24]. Reversine alone severely suppressed the number of cells in mitosis (Figure 2H), caused the formation of multi-nucleated cells (Appendix A) and reduced the number of viable cells (Appendix A), the latter suggesting that post mitotic apoptosis was triggered. Reversine was able to override the vincristine induced mitotic arrest, and this was accompanied by the appearance of multinucleated cells (Figure 2H and Appendix A). Most importantly, DEXA also reduced the number of cells in vincristine-induced mitotic arrest; it induced the formation of multinucleated cells and increased the number of cells that survived vincristine treatment (Figure 2H–J). At the molecular level we found that the localisation of BUB1 and CENPF at kinetochores was reduced in the presence of DEXA (Appendix A) further corroborating that DEXA compromises the SAC.

The ability of DEXA to override the SAC would suggest that it is able to support the continued division of cells, even if they have encountered radiation-induced DNA damage. Indeed, confirming our observations made in the screen with T98G cells (Figure 1), DEXA was stimulating cell division in several GBM cell lines even when they had been radiated (Figure 2E and Appendix A). Most importantly however and in line with the ability of DEXA to override the checkpoint that produces a mitotic arrest, its positive effect on the propagation of radiated cells also occurred when administered sometime after cells had been irradiated (Figure 2K).

### 2.4. Dexamethasone Up-Egulates PDGFR Mediated Survival Signalling 

Overriding the SAC allows DNA damage to be translated into chromosome abnormalities, but for continued proliferation with chromosome abnormalities appropriate survival signalling is required [20,22].

To identify what enables GBM cells to continue to proliferate in the presence of DEXA, we analysed our T98G RNAseq-data. This revealed significant enrichment of receptor tyrosine kinase (RTK) and particularly PDGFR signalling in DEXA treated cells (Figure 3A). Indeed, we observed a modest up-regulation of components of the PDGFR signalling module, including the ligand PDGFB in response to DEXA, although this varied amongst different GBM cell lines (Figure 3B,C). Moreover, treatment with DEXA led to an increase in the auto-phosphorylation of PDGFRA and PDGFRB (Figure 3D), which can occur in homo- and heterodimers [25]. Phosphorylation of Y572 in PDGFRA and/or Y579 in PDGFRB creates binding sites for SRC family kinases (SFKs) including SRC, FYN and YES as well as STAT5 [25] and we found phosphorylation (i.e. activation) of both after 18 h of DEXA treatment in T98G and A172 cells (Figure 3E).

The activation of SRC and STAT signalling downstream of RTKs can induce survival signalling through BCL2L1 or MCL1 [26,27]. In fact, STAT5B can be activated by SFKs and mediate survival signalling in glioblastoma cells through direct activation of the *BCL2L1* promoter [28]. Moreover, STAT5B can drive tumour progression in a PDGFB-driven glioma model and this involves increased BCL2L1 expression [29]. In the TCGA GBM patient cohort [4] STAT5B and the expression of SRC, FYN and YES are strongly correlated with the expression of BCL2L1, but also with MCL1 (Figure 3F and Appendix A). Importantly, PDGFRB expression is also significantly correlated with STAT5B, BCL2L1 and MCL1 expression (Figure 3F and Appendix A). In line with our previous observations, DEXA stimulated the mRNA expression of both, BCL2L1 and MCL1 in T98G and A172 cells (Figure 3G). In SF118 and LN229 cells it only induced MCL1 or BCL2L1, respectively (Figure 3G), which was correlated with inefficient activation of SRC and STAT5 (not shown). In T98G and A172 cells, the DEXA induced BCL2L1 and MCL1 expression was reduced by the potent PDGFRB inhibitor ponatinib (Figure 3H), demonstrating that ponatinib can block DEXA-induced signalling.

Inactivation of BCL2L1 and MCL1 during prolonged mitosis has been considered the priming event in mitotic death signalling [20]. With DEXA reducing mitotic control through the SAC, increased expression of BCL2L1 and MCL1 could enable cells to survive with mitotic errors. As such, the balance between lowering the activity of the SAC and enhancing survival signalling could contribute to the net effect of DEXA on GBM cell growth, which indeed is notoriously variable amongst different GBM cell lines [13]. We found that DEXA induces colony formation in T98G and A172 cells, in which both BCL2L1 and MCL1 are up-regulated, but no significant effect was seen in SF188 cells and DEXA was inhibitory in LN-229 cells (Figure 3I).

The relevance of PDGFR signalling for long-term growth was corroborated by the fact that the DEXA-induced colony formation of T98G and A172 cells was suppressed by ponatinib and another PDGFR inhibitor, sunitinib (Figure 3J). Moreover, the DEXA-induced growth effect was also abolished by the MCL1 inhibitor S63845 and the pan BCL2 family inhibitor navitoclax, which also inhibits BCL2L1 (Figure 3K). Furthermore, no inhibition was seen with the BCL2 specific inhibitor venetoclax (Figure 3K), emphasizing the specific role of BCL2L1 in DEXA mediated survival signalling.

### 2.5. PDGFR Expression and SRC Kinase Activation Correlate in High-Grade Gliomas

We next assessed whether the pathway relevant for DEXA-induced long-term growth was activated in human glioblastoma. We focused on SFKs, because reliable antibodies against the activated forms (phospho-SFK) are available. As seen for STAT5B (Appendix A), both PDGFRA and PDGFRB significantly correlate with the expression of SRC, FYN and YES in the TCGA patient cohort (Figure 4A and Appendix A). Analysis of a tissue microarray (TMA) identified both PDGFRA expression and basal SFK phosphorylation as detectable in the majority of normal glial cells in healthy brain tissue as well as in GBM tumours (Figure 4B). On the other hand, whereas PDGFRB expression was only seen in 22% of normal glial cells, in GBM tumours its expression was increased and detectable in 65% (Figure 4B). PDGFRA and PDGFRB displayed high expression in 59% and 22% of samples, respectively (Figure 4C,D), whereas high SFK phosphorylation was seen in approximately half of all tumours (Figure 4E). In these “phospho-SFK high” tumours 66% also expressed high PDGFRA or PDGFRB or both (Figure 4F), supporting a scenario in which PDGFR signalling contributes to SRC/SFK activation in high-grade gliomas.

### 2.6. Tyrosine Kinase Inhibition Overcomes Dexamethasone-Mediated Radioprotection 

We next wished to analyse the impact of DEXA-induced PDGFR signalling on radiated cells. Intriguingly, radiation alone led to up-regulation of PDGFR, its ligand PDGFB, BCL2L1 and MCL1, albeit with some variations (Figure 5A). 

The effect of radiation on mitosis-control genes was diverse with up-regulation in T98G cells and strong down-regulation in A172 cells (Figure 5A). DEXA generally induced a similar profile in transcriptional changes to what we had seen in non-radiated cells with up-regulation of PDGFRs, BCL2L1 and MCL1 but down-regulation of mitosis-control genes (Figure 5A). As with non-radiated cells, this impacts on the balance between reduced mitosis control and increased survival signalling. Accordingly, the radioprotection capacity of DEXA varied with radio-protective effects seen in T98G, SF188 and LN229 cells but an inhibitory effect in A172 cells, in which DEXA strongly down-regulated mitosis and cell division related genes (Figure 5A,B).

As we had observed up-regulation of PDGFR and its ligand, we assessed the radio-protective capacity of PDGFR signalling in the absence of DEXA by exposing GBM cells to recombinant PDGFB. We found that PDGFB was not only generally pro-proliferative but also protects radiated cells (Figure 5C), supporting the relevance of PDGFR signalling in the context of radiation. Intriguingly, in radiated SF188 cells, the effect of PDGFB alone was much weaker than what we had previously observed with DEXA (Figure 5B). We therefore pre-treated radiated SF188 cells with DEXA and this significantly increased the radio-protective effect of PDGFB (Figure 5D), further supporting the idea that DEXA “primes” glioblastoma cells towards PDGFR mediated growth and survival signalling.

To analyse whether inhibiting PDGFR signalling could overcome the radio-protective effect of DEXA, we used the tyrosine kinase inhibitors (TKIs) ponatinib and sunitinib, which have a high affinity for PDGFR, as well as dasatinib, which also inhibits SFKs. All three inhibitors were effective in GBM cells (Appendix A) and significantly inhibited GBM cell colony formation after radiation, even when DEXA provided a radio-protective effect (Figure 5E). 

### 2.7. Dexamethasone Sensitizes Glioblastoma Cells to Sunitinib 

In some radiated GBM cell lines, we saw an enhanced response to ponatinib and dasatinib in the presence of DEXA (Figure 5E). However, with sunitinib a significant increase in the inhibitory effect was seen in all cell lines when DEXA was present (Figure 6A). Moreover, this effect was independent of radiation and was also observed in non-radiated cells (Figure 6A), which suggested that DEXA induced signalling generally sensitizes GBM cell lines to sunitinib. 

While GBM cell lines provide vital information with regard to drug responses, it is GSCs that are thought to be the source of therapeutic resistance and tumour recurrence after surgery, and they are considered a critical target in a successful therapy approach [30]. When we analysed GSCs derived from mice overexpressing PDGFA but lacking TP53 and NF1, we found that independent of radiation the basal neurosphere formation capacity was not sensitive to sunitinib (Figure 6B,C). 

However, DEXA, which was not only promoting neurosphere formation in non-radiated and radiated GSCs, also profoundly sensitized these GSCs to the inhibitory effect of sunitinib (Figure 6B,C). This further supports the idea that DEXA induced signalling establishes a vulnerability for sunitinib in glioblastoma cells.

DEXA reduces the development of cerebral edema in GBM patients by blocking the production of pro-inflammatory cytokines, which otherwise affect blood-brain-barrier (BBB) functionality. As microglial cells and astrocytes play an important role in this scenario, we wished to examine the effect of the DEXA/sunitinib combination treatment on these cells. DEXA suppressed the expression of the pro-inflammatory cytokines IL1B and TNFA in LPS/IFNγ activated microglial cells, and sunitinib increased this response (Appendix A). A similar situation occurred with IL10 (Appendix A), but no effect was seen with the marker of the M2a alternative activated phenotype, ARG1 (Appendix A). In astrocytes, the DEXA/sunitinib combination suppressed LPS/IFNγ-induced IL1β but not TNFα expression (Appendix A). The expression of IL10 and ARG-1, both suppressed by LPS/IFNγ, was recovered by DEXA/sunitinib treatment.

### 2.8. Dexamethasone Induces a Therapeutic Vulnerability for Sunitinib In Vivo 

To assess whether the DEXA induced vulnerability occurs in vivo, we used U251MG cells, because they are relatively resistant to sunitinib, dasatinib and ponatinib (see Appendix A), but display significant synergy when treated with DEXA and sunitinib (see Figure 6A). Similar to what is seen in radiated cells (see Figure 5A) DEXA treatment of non-radiated U251MG cells also suppresses PLK1, KIF11 and PTTG1 expression, but TTK expression is induced by DEXA in these cells (Appendix A). BCL2L1 and MCL1 expression are induced in U251MG cells by DEXA, which correlates with a pro-proliferative effect in vitro (Appendix A).

For the in vivo treatment, we chose a dose of 0.3 mg/kg DEXA once daily in order to minimize effects on weight loss (Appendix A) as this effect had been described previously [31,32]. At this dose DEXA maintained tumour growth, but we did not observe a significantly increased mean tumour volume (Figure 7A). This suggested that we either did not reach concentrations high enough to detect the pro-proliferative effect, or that DEXA-induced effects on the tumour microenvironment counteracted proliferation. Importantly however, under conditions where sunitinib was entirely ineffective in reducing tumour volume, the additional presence of DEXA led to a significant reduction in tumour growth (Figure 7A), demonstrating that the sensitization of GBM cells by DEXA observed in vitro also occurs in vivo. 

We used phospho-SRC/SFK to monitor effects on downstream signalling activated by DEXA. Basal phosphorylation in control mice varied from very low to medium intensity (Figure 7B, Appendix A). DEXA induced an increase in phospho-SRC signal, but strikingly this was also seen with sunitinib (Figure 7B and Appendix A). Such an increase in phosphorylation has been observed previously with imatinib and ponatinib and was suggested to be based on compensatory signalling induced by these TKIs when used as single agents at low/ineffective concentrations [33,34]. Importantly however, when used in combination with DEXA sunitinib led to a striking reduction in phospho-SRC signal (Figure 7B and Appendix A). 

Despite the increase in phospho-SRC signal in DEXA treated tumours, we did not detect a significant increase in BCL2L1 and MCL1 expression (Figure 7C,D). This suggests that the DEXA-induced signaling was only sufficient to maintain the expression of survival signalling, which is in line with the lack of increased growth. Nevertheless, although not reaching significance, there was a trend of reduced expression of PLK1, KIF11, PTTG1 and TTK (the latter otherwise induced by DEXA in U25MG cells (see Appendix A) in DEXA treated tumours (Figure 7E–H). More importantly, the DEXA/sunitinib combination induced a significant decrease in mRNA expression of survival and mitosis-control genes (Figure 7C–H). Thus, while DEXA alone did not produce significant changes in all assessed parameters, adding sunitinib to DEXA treated tumours led to a significant repression of survival and mitosis-control genes, and reduction in tumour growth. This suggests that DEXA had primed the tumour cells to signal in a sunitinib-dependent manner.

In summary, we identified a mechanism in which DEXA compromises the SAC by reducing the amount of relevant players such as PLK1, MPS1, BUB1 and CENPF, and concomitantly promotes PDGFR signalling, which contributes to survival (see Figure 7I). As such, DEXA drives GBM cell growth into dependency of signalling that can be inhibited by sunitinib.

## 3. Discussion

Dexamethasone is the corticosteroid of choice used in the management of cerebral edema in GBM treatment, but several clinical studies found that low steroid use during radio/chemotherapy correlated with better survival [8,9,10,11,12]. Similar observations were made in a PDGFB-driven mouse model [18] and the interference of DEXA with the microenvironment including the immune-microenvironment has been linked to its negative impact on radiotherapy [12,18,35,36].

We show here that DEXA can have a direct radio-protective effect on human GBM cells as well as GSCs in vitro. However, this did not happen in all GBM cell lines, and in fact in A172 cells DEXA sensitised to radiation. This variability is reminiscent of the fact that DEXA can be pro-proliferative or anti-proliferative in glioblastoma cells in vitro, an effect we did observe and that has been described previously [13]. Even in vivo, DEXA was found to be anti-proliferative in PDGFB-driven gliomas [18], but to be pro-proliferative in GSC-derived orthotopic xenografts [37]. One explanation for this variability could be that the cellular background i.e. the genetic make-up of individual glioblastoma cells impacts on the DEXA induced transcriptional changes and signalling.

Nevertheless, we found that independently of the mutation/amplification/deletion status of major GBM classifiers such as EGFR, PDGFRA or NF1, DEXA consistently suppressed G2/M transition/SAC related genes in GBM cell lines as well as in GSCs, and the same trend was observed in radiated cells. We identified a mitosis-control gene signature similar to Pitter and co-workers [18] and low expression of these signature genes correlated with poor prognosis in GBM. Intriguingly, in lower grade glioma and in other cancers such as lung, colon and breast cancer (not shown) low expression of these signature genes correlates with better prognosis, implying that the response to reduced mitosis-control is distinct in GBM when compared to other cancer types.

We found that DEXA can propagate GBM cell growth even after radiation, suggesting that the ability of DEXA to override the SAC allows for continued proliferation despite DNA damage. However, for continuous proliferation to occur cells also need to activate survival signalling [20].

After radiation p53-dependent post-mitotic responses induce cell cycle arrest, followed by apoptosis or senescence [38], but apart from A172 in which DEXA sensitized cells to radiation, the glioma cell lines we used express mutated TP53. In addition, DEXA up-regulated BCL2L1 and MCL1, the two major regulators of survival during an extended mitotic arrest as well as in post-mitotic apoptosis [20]. Importantly, this up-regulation varied amongst the cell lines and might not have always been strong enough to warrant survival. Nevertheless, in A172 cells BCL2L1 and MCL1 were induced, yet DEXA sensitised the cells to radiation. This could be partly due to the fact that p53 activity is maintained in A172 cells, however DEXA also severely suppressed mitosis regulation genes in this line. Thus, an appealing hypothesis emanating from our data is that the balance between down-regulation of mitosis-regulators and up-regulation of BCL2L1 and MCL1 is linked to the pro-proliferative and radio-protective activities of DEXA. Addressing this hypothesis however remains subject of future studies.

While DEXA suppressed mitosis control it concomitantly up-regulated PDGFR signalling in GBM cells. PDGFR inhibition in GBM cell lines universally triggers a G2/M arrest [39] and in a PDGFRA/PDGFA driven glioblastoma mouse model chronic activation of PDGFRA facilitates microtubule dynamics during mitosis [40], suggesting a role for PDGFR in driving GBM cells towards mitosis. Thus, the DEXA induced activation of PDGFR signalling is in line with the increase of cells in mitosis we observed.

We detected phosphorylation of SFKs and STAT5, both acting downstream of PDGFRs [25], in response to DEXA. STAT5B-phosphorylation has been linked to poor survival in GBM, and in an EGFRvIII-driven GBM mouse model SFK mediated STAT5B activation regulates expression of AURKA and BCL2L1 [28], expression of both is induced by DEXA.

We demonstrate that using TKIs not only overcomes DEXA mediated pro-proliferative and radio-protective activities, but also that DEXA sensitises GBM cells and GSCs to sunitinib. DEXA supports GSCs neuro-sphere formation of radiated GSCs, suggesting that DEXA can support resistance to radiotherapy in GBM. Crucially, sunitinib can overcome the DEXA mediated radiotherapy resistance in GSCs, where it also profoundly sensitized the cells to the TKI. Moreover, we detected a similar sensitisation by DEXA to sunitinib in vivo, further emphasizing our finding that DEXA creates a therapeutic vulnerability for sunitinib in GBM. 

Previous trials assessing sunitinib in GBM patients have been hampered by limitations such as small sample size and vast heterogeneity in prior therapy [41,42,43,44,45]. Furthermore, while restriction for BBB crossing has been discussed, sunitinib has in fact shown activity in the brain in a large trial involving 321 renal cell carcinoma patients with brain metastasis [46], suggesting that BBB crossing should not pose a challenge. Nevertheless, our data imply that in GBM the presence of DEXA impacts on the efficacy of sunitinib, and this factor, which was not considered in previous trials, might have added an unpredicted variability to respective study outcomes.

Our findings suggest that through transcriptional rewiring of glioblastoma cells, DEXA creates a therapeutic vulnerability for tyrosine kinase inhibitors in GBM that could be exploited in future therapy approaches. We believe there is still ground for using multi-targeted RTK inhibitors for the treatment of GBM, and we propose that with their DEXA-ameliorating activity, such RTK inhibitors could be used in a post-surgical adjuvant setting with or without concomitant radiotherapy. 

## 4. Material and Methods

### 4.1. Patients 

In total, 285 patients seen at the Donostia University Hospital, San Sebastian and diagnosed with primary glioblastoma grade IV according to the World Health Organisation criteria were included in the study. All participants signed informed consent forms approved by the Institutional Ethical Committee (ethical code: PI2016151, 22/02/2017). The study was approved by the ethic committee of the Biodonostia Institute and the Donostia University Hospital.

### 4.2. Cell Culture and Reagents

T98G, A172, LN229, MCF-10A and HMC3 human microglial cells were from ATCC; U251-MG and SF188 cells were a gift from Dr Chris Jones (ICR, London, UK). GBM cell lines were cultured in Dulbecco’s Modified Eagle’s Medium (DMEM) (cat#11995065, Gibco), HMC3 cells in DMEM/F12 (cat#10565018, Gibco), and MCF-10A cells in mammary epithelial cell basal medium (MEBM) (cat#CC-3151, Lonza). Human astrocytes (cat#3P10251; Innoprot) were cultured in AM (cat#1801, ScienCell) with supplements (cat#1852, ScienCell). Mouse GSCs were isolated and cultured as described previously [47]. For radiation, cells were irradiated with a linear electron accelerator (Varian Medical Systems) using a 6MV photon field at varying doses ranging from 6−12 Grays. Dexamethasone (cat#D4920), temozolomide (cat#T2577), human PDGF-BB (cat#P3201) and LPS (cat# L5293) were from Sigma/Merck (Madrid, Spain). IFNγ (cat#300-02) was from Peprotech (London, UK) and reversine (cat#ab120921) from abcam (Cambridge, UK). FDA approved drug library (cat# L1300), Ponatinib (cat#S1490), Sunitinib (cat#S7781), Dasatinib (cat# S1021), Venetoclax/ABT-199 (cat# S8048), S63845 (cat# S8383), Vincristine (cat#S1241) and Taxol (cat#S1150) were from Selleckchem (Newmarket, UK). Navitoclax/ABT-263 (cat#M1637) was from AbMole. TKIs were used at concentrations in the range of their IC50s (see Appendix A).

### 4.3. FDA-Approved Drug Screen

A library with 978 FDA approved drugs (Z148990, Selleckhem) was used for the drug screen. T98G cells (cat# CRL-1690) were plated in 96-well format, and 24 h after plating and 3 h before a 12 Grey single dose radiation, cells were treated with the drugs at 10 µM. For the analysis, crystal violet staining was used, because it provided reliable measurements of cell numbers after radiation in a dose dependent manner. At confluence of the control, cells were fixed with 4% paraformaldehyde in phosphate buffered saline (PBS) and stained with freshly prepared 0.1% crystal violet. Following rinsing with distilled water the stained cells were dissolved in 200 µl 1% SDS and absorbance at 570 nM was measured against a control. Twenty-six compounds with significant protection were selected in the first screen, and re-screened at concentrations of 1, 10 and 25 µM using 6 Gy. The second screen identified 13 glucocorticoids (listed in Figure 1B) for their capacity to protect from radiation-induced growth inhibition.

### 4.4. RNA Analysis 

For RNAseq of T98G cells, RNA from cells either left untreated or treated with DEXA for 18 h was extracted using an RNeasy Mini Kit (Qiagen 74104). Total RNA was processed as previously described [48]. The sequencing run was performed on an Illumina HiSeq1500 instrument. The quality of the raw unprocessed reads was evaluated using the FastQC software. The reference genome and the reference annotation were obtained from the Ensembl database. Clean reads were aligned to reference human genome (GRCh38) using the STAR aligner in the 2-pass mode. FeatureCounts was used to generate counts of uniquely mapped reads to annotated genes using the reference annotation (version 92) file. Differential gene expression analysis was performed using R-Package DESeq2 with a threshold *p*-value < 0.05 after false-discovery rate correction. The fold expression of genes with padj < 0.05 is shown in Appendix A. Functionality analysis was performed using Metascape [49]. For qRT-PCR experiments, total RNA was extracted and analysed as described previously [50]. Primer sequences are provided in Appendix A.

### 4.5. Colony Formation Assay 

Cells seeded in 6-well plates were treated with DEXA (cat#D4902, Sigma) as indicated. If present, inhibitors or DMSO were added 18 h after DEXA addition and 3 h before radiation. When control cells had reached density, cells were analysed as described previously [51]

### 4.6. Neurosphere Formation Assays

Neurosphere formation of KAB-194 mouse GSCs was performed as previously described [52]. DEXA was added 8 h after disaggregation and reseeding. Inhibitors were added 18 h after DEXA addition and 3 hours before radiation. 

### 4.7. Analysis of Mitotic Errors, SAC Override and Kinetochore Localisation

Cells on coverslips were treated with DEXA for 48 h, fixed (4% paraformaldehyde) and stained with Hoechst 33258 (cat#B2883, Sigma). In total, 500 cells were counted and the number of cells in mitosis reported. For quantification of mitotic errors, 50 mitotic cells were counted, and the number of mitotic errors reported. For α-tubulin staining, fixed cells were washed with PBS/0.2% Triton X-100 and incubated with anti-α-tubulin (DM1A, cat#05-829, Merck) for 2 h, washed three times and incubated with Alexa-Fluor488 anti-mouse (A-11001, Thermo Fisher, Carlsbad, CA, USA) for 1 h. For analysis of SAC override, cells were stained with anti-phospho-H3/S10 (cat#ab14955, Abcam) as described above and 300 cells per condition were analysed. For the kinetochore localization analysis anti-CREST, anti-BUB1 and anti-CENPF [53] were used. Cells were imaged using a Leica DM4000 microscope and analysed with NIS-Elements software (Nikon, Amsterdam, Netherlands).

### 4.8. Tissue Microarray Analysis

Tissue microarrays were purchased from US Biomax Inc. (Rockville, MD, USA, cat# GL805e). Immunohistochemistry was performed using the Leica Bond Max system (Leica Biosystems, Wetzlar, Germany) and the Bond Polymer Refine Detection Kit (DS9800, Leica). TMA sections were de-paraffinised and rehydrated in decreasing alcohol concentrations. Endogenous peroxide activity was quenched using Peroxide Block for 15 min and tissue sections were subjected to heat-induced antigen retrieval in a steamer (98 °C for 20 min), using Bond Epitope Retrieval Solution 1 (AR 9961, Leica) (pH 6.0). Tissue sections were incubated for 60 min with the respective primary antibodies. Post Primary solution was added for 8 min and Polymer solution for another 8 min. Then the chromogen DAB (3.3-diaminobenzidine-tetrahydrochloride; Sigma, St Louis, MA, USA) was applied at room temperature (RT) for 10 min and Hematoxylin for 5 min. Primary antibodies were for: phospho-SRC (Tyr419) (#44-660G, Invitrogen, 1:500), PDGFR-α (Rb-9027, ThermoFisher, 1:200), and PDGFR-β (#DPABH-01589; Creative Diagnostics, 1:500). Slides were evaluated by at least two of the authors (P.A. and I.A.) and then submitted to an independent experienced pathologist (M.V.Z.) for the final score. The staining for each core was determined as negative, low and high.

### 4.9. Cell Lysis and Western Blot Analysis

Cells were lysed and analysed by Western blotting as described previously [51]. Primary antibodies were: PDGFRα (cat# 3174S), PDGFRβ (cat# 3169S), p-PDGFRα (Y849)/β(Y857) (cat# 3170T), p-PDGFβ (Y740) (cat# 3168), p-SRC (Y416) (cat# 2101) and p-STAT5 (cat# 4322T) from Cell Signaling; p-PDGFRα (Tyr572)/PDGFRβ (Tyr579) (cat# bs-5554R) from Bioss Inc and β-Actin (cat# A5441) from Sigma. Detection was through enhanced chemiluminescence ECL using Horse Radish Peroxidase (HRP)-coupled secondary antibodies (GE Healthcare) and NOVEX ECL Chemi Substrate (ThermoFisher).

### 4.10. In Vivo Drug Treatment

All processes involving animals were subject to approval by the Biodonostia HRI animal experimentation ethics committee. In total, 2 × 10^6^ U251-MG cells were injected into both flanks of Foxn1^nu^/Foxn1^nu^ nude mice (8 weeks of age). External callipers were used to measure tumour volume. Once tumours head reached size, mice were assigned to different groups (*n* = 7 per group) with an average tumour volume of ~25 mm^3^. Drugs or vehicle were administered by intraperitoneal injection (IP). Vehicle, DEXA (0.3 mg/kg), sunitinib (40 mg/kg) or a combination was administered once daily for 10 days and tumour volumes measured on day 3, 6 and 9. Tumours were collected and either paraffin-embedded for immunohistochemistry or snap-frozen for RNA extraction.

### 4.11. Data Analysis and Statistics

GraphPad Prism version 7.00 for Mac OS (GraphPad Software, San Diego, CA, USA) was used for analysis. One-way ANOVA or Student’s t test was used for bar graph analyses, log-rank test for Kaplan–Meier survival analyses, Pearson correlation for co-expression analyses and two-way ANOVA (mixed model) analysis for tumour growth. Data represent the results for assays performed from at least 3 replicates, and values are the mean ± SEM. * *p* < 0.05; ** *p* < 0.01; *** *p* < 0.001.

## 5. Conclusions

When DEXA is administered during radiotherapy in GBM patients this correlates with reduced overall- and progression-free survival. Our data suggest that DEXA can directly protect GBM cells from radiation by compromising the SAC and concomitantly increasing survival signalling. The SAC is one of the most crucial cell cycle checkpoints hindering cells with DNA/chromosome damage to divide. We demonstrate that DEXA can propagate GBM cell growth even some time after radiation, suggesting that the ability of DEXA to override the SAC promotes continued proliferation despite DNA damage. In this scenario the DEXA-induced PDGFR/survival signalling may increase the threshold for mitotic catastrophe to set in and may enable GBM cells to better adapt to genomic abnormalities. This suggests a crucial dependence on PDGFR signalling and we show that TKIs overcome the DEXA mediated pro-proliferative and radio-protective activities, and that furthermore DEXA sensitises GBM cells and GSCs to sunitinib. The novel vulnerability that we reveal encourages the revision of the use of TKIs in future trials, whereby not only the level of DEXA use is monitored but also TKI use is considered in an adjuvant setting.

## Figures and Tables

**Figure 1 cancers-13-00361-f001:**
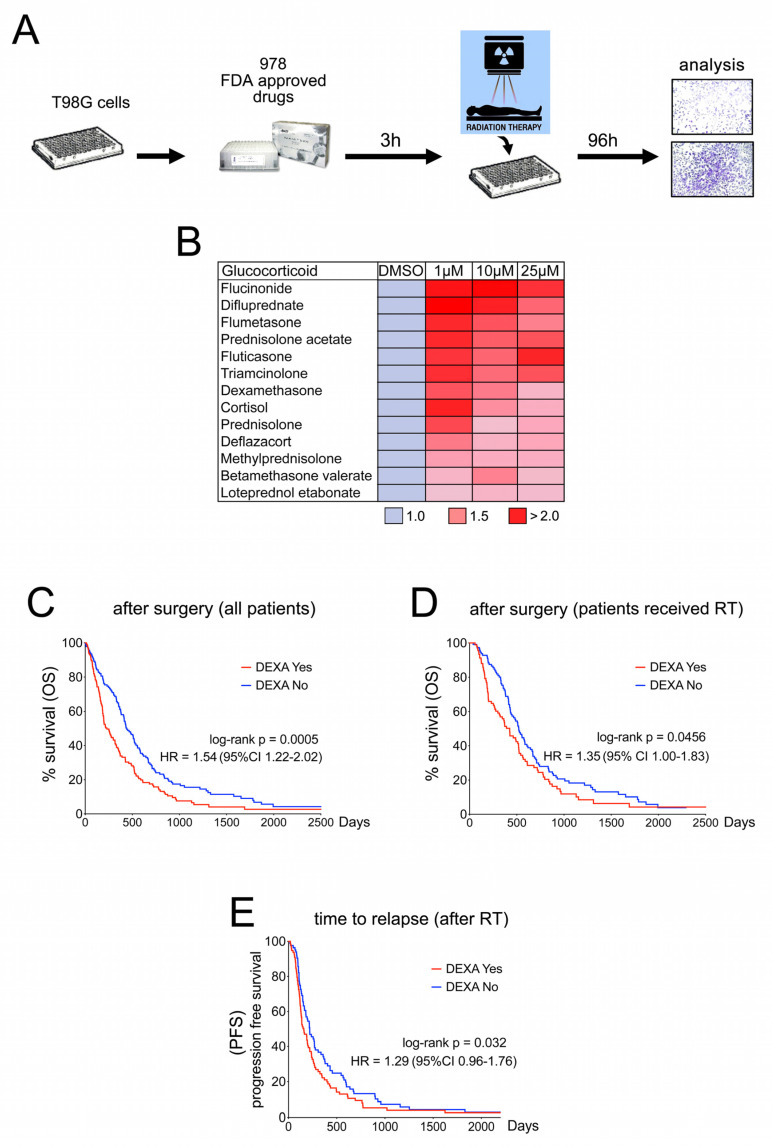
Dexamethasone protects from radiotherapy and reduces survival in glioblastoma (GBM) patients. (**A**) Schematic showing workflow of the FDA approved drug screen. T98G cells were treated with a library of 978 FDA approved drugs at 10 µM, exposed to a single dose of 12 Gy and 96 h later analysed for survival using crystal violet staining. (**B**) 26 drugs selected in the first screen were re-screened at concentrations of 1, 10 and 25 µM using 6 Gy. Thirteen glucocorticoids identified in this second screen are shown. Dimethyl sulfoxide (DMSO) was set = 1. (**C**) Kaplan-Meier analysis of the Biodonostia patient cohort. Differences in overall survival for patients receiving no corticosteroids (*n* = 141) or dexamethasone (*n* = 144) are shown. Hazard ratio and p (log-rank) are indicated. (**D**) Kaplan-Meier analysis of the Biodonostia cohort of patients who underwent radiotherapy (*n* = 207). Differences in overall survival for patients receiving basal (*n* = 93) or no (*n* = 114) dexamethasone are shown. Hazard ratio and p (log-rank) are indicated. (**E**) Kaplan-Meier analysis for progression free survival after radiotherapy for patients receiving no (*n* = 114) or basal dexamethasone (*n* = 93). Hazard ratio and p (log-rank) are indicated.

**Figure 2 cancers-13-00361-f002:**
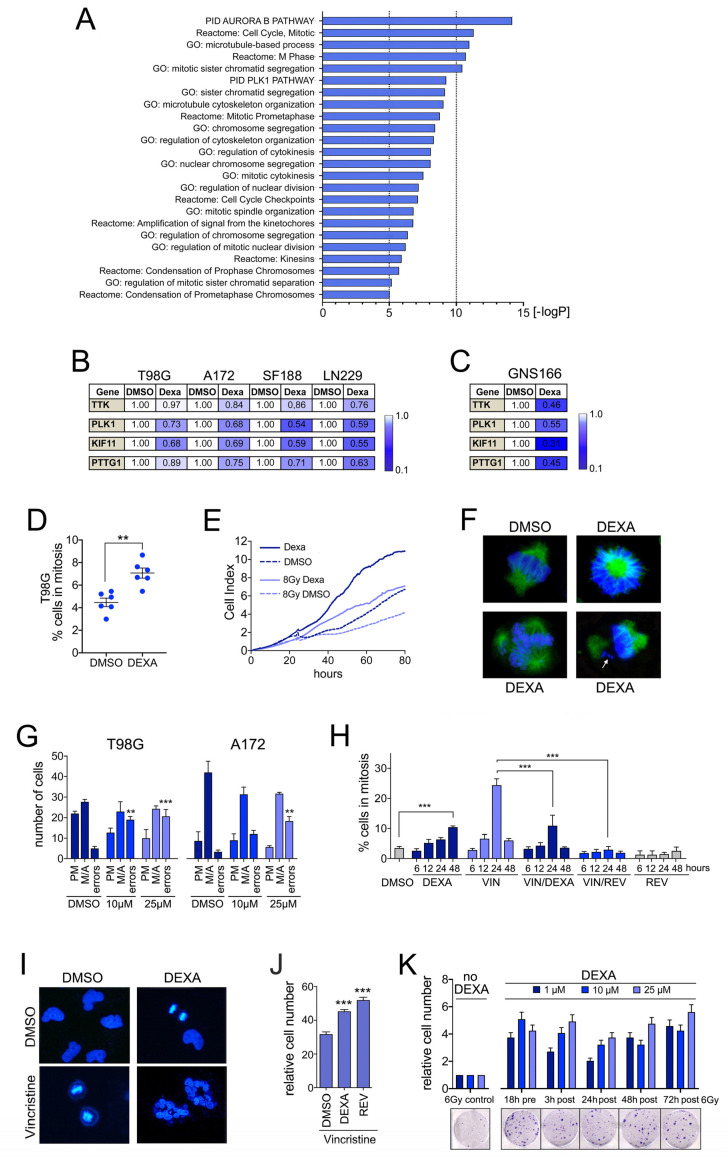
Dexamethasone (DEXA) suppresses mitosis control genes and overrides the spindle assembly checkpoint (SAC). (**A**) Functional characteristics of T98G cells treated with 25 µM DEXA for 18 h revealed by Gene Set Enrichment Analysis (GSEA) using Gene Ontology (GO) term, Reactome and Pathway Interaction Dataset (PID) gene set collections. (**B**) qRT-PCR analysis for TKK, PLK1, KIF11 and PTTG1 in the indicated cell lines represented as mean fold change of triplicates treated with DEXA for 18 h relative to DMSO. (**C**) qRT-PCR analysis for the indicated genes in GNS166 cells as in (**B**). (**D**) % T98G cells in mitosis, quantified (*n* = 3 experiments) 48 h after addition of 25 µM DEXA. (**E**) iCELLigence™ proliferation analysis of T98G cells either non-radiated or radiated with 8G y in the absence or presence of 10 µM DEXA. (**F**) T98G cells either untreated (DMSO) or treated with 25 µM DEXA for 48 h were analysed with anti-α-tubulin and stained with Hoechst 33258 and imaged. (**G**) Mitotic errors (monopolar or tripolar spindle, lagging chromosomes, chromosome bridges) per 50 mitotic cells were quantified. Data represent the mean ± SEM (*n* ≥ 3). (**H**) T98G cells treated with 5 nM vincristine with or without 25 µM DEXA were stained for phospho-H3 and quantified at the indicated times. Treatment with 1 µM reversine (REV) served as control. (**I**) T98G cells treated with 5 nM vincristine with or without 25 µM DEXA for 48 h were stained with Hoechst 33258 and imaged. (**J**) T98G cells were treated as in (**H**) and analysed for colony formation (mean ± SEM (*n* ≥ 3); DMSO treated cells were set 100% and 1 µM reversine (REV) served as positive control. (**K**) Cell number quantification of T98G cells radiated with 6 Gy either in the absence (control) or in the presence of DEXA. Thus, 10 µM DEXA was added to the cells either prior (pre) or after (post) radiation as indicated. Control cells were set 1, *n* = 3. Images represent examples. ** *p* < 0.01, *** *p* < 0.001.

**Figure 3 cancers-13-00361-f003:**
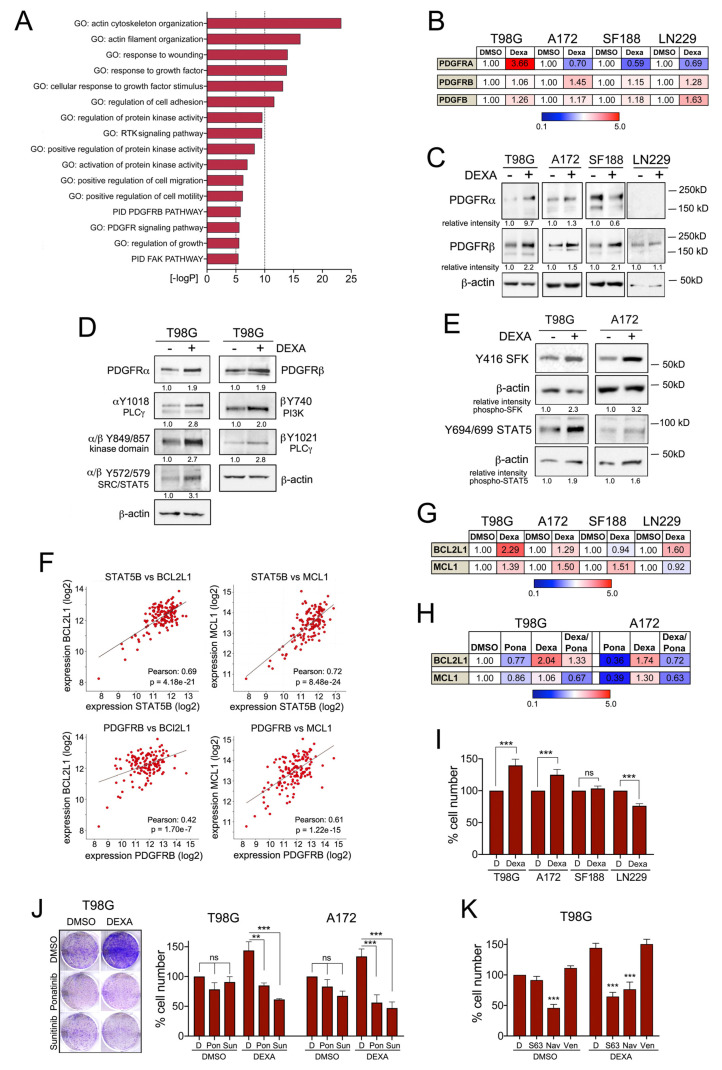
Dexamethasone induces PDGFR mediated survival signalling. (**A**) Functional characteristics of T98G cells treated with 25 µM DEXA for 18 h revealed by GSEA using GO term and PID gene set collections. (**B**) qRT-PCR analysis for PDGFRA, PDGFRB and PDGFB in the indicated cell lines represented as mean fold change of triplicates treated with DEXA for 18 h relative to DMSO. (**C**) Western blot analysis for PDGFRA and PDGFRB in the indicated cell lines either untreated or treated with DEXA for 18 h. Beta-actin served as loading control. (**D**) Western blot analysis for PDGFRA, PDGFRB and the indicated phospho-specific antibodies in T98G cells either untreated or treated with DEXA for 18 h. The lysate of one experiment was run on several lanes per blot, which were probed in parallel. Beta-actin serves as loading control and representative controls are shown. (**E**) Western blot analysis for phospho-SFK (using a phospho SRC antibody) and phospho-STAT5 in the indicated cell lines either untreated or treated with DEXA for 18 h. Beta-actin served as loading control. (**F**) Co-expression analysis of the indicated genes from the TCGA Glioblastoma dataset. (**G**) qRT-PCR analysis for BCL2L1 and MCL1 in the indicated cell lines represented as mean fold change of triplicates treated with DEXA for 18 h relative to DMSO. (**H**) qRT-PCR analysis as in (**G**). Cells were treated with DEXA in the absence or presence of 200 nM ponatinib for 18 h. (**I**) Quantification of the relative cell number of the indicated cell lines grown in the absence or presence of 25 µM DEXA. Data represent the mean ± SEM (*n* ≥ 3). (**J**) Quantification of the relative cell number of the indicated cell lines grown with or without 25 µM DEXA in the absence or presence of 25 nM or 100 nM ponatinib (Pon) or 1 µM or 3 µM sunitinib (Sun). Data represent mean ± SEM (*n* ≥ 3). (**K**) Quantification of the relative cell number of T98G cells grown with or without 25 µM DEXA in the absence or presence of 5 µM S63845 (S63), 1 µM navitoclax (Nav) or 5 µM venetoclax (Ven). Data represent mean ± SEM (*n* ≥ 3). ** *p* < 0.01, *** *p* < 0.001.

**Figure 4 cancers-13-00361-f004:**
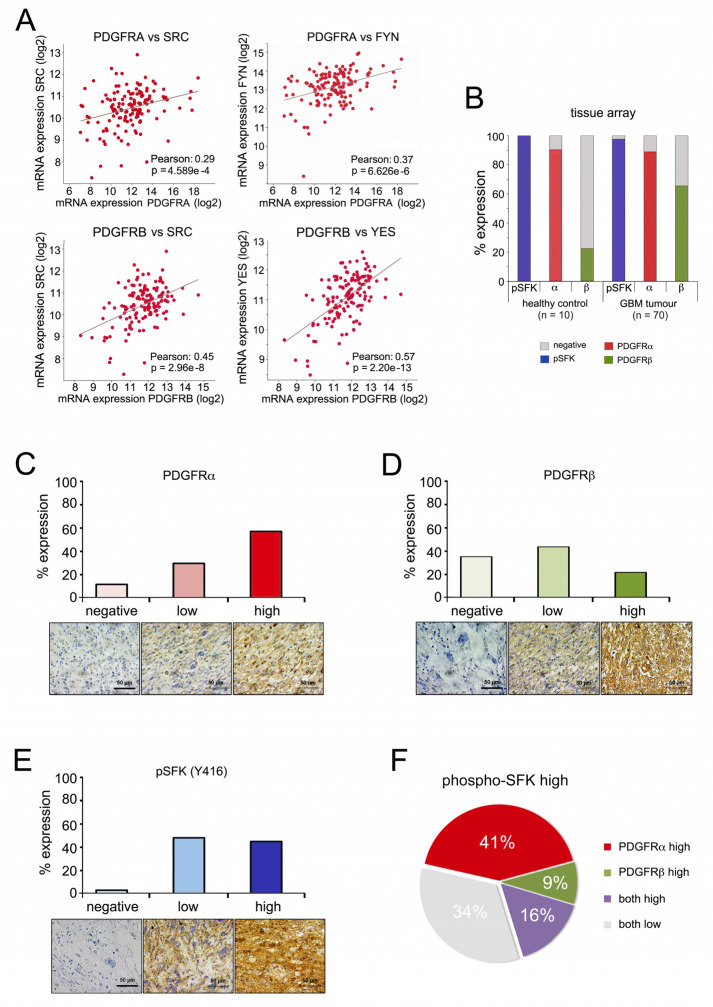
PDGFR and SRC kinase co-expression in high-grade gliomas. (**A**) Co-expression analysis of the indicated genes from the TCGA Glioblastoma dataset. (**B**) Overall expression of pSKF (using a phospho-SRC antibody), PDGFRA and PDGFRB in a TMA containing 70 stage 4 Glioblastoma and 10 healthy cerebrum tissue samples. (**C**) Quantification of relative PDGFRA expression in TMA GBM samples; scale bar, 50 µm. (**D**) Quantification of relative PDGFRB expression in TMA GBM samples; scale bar, 50 µm. (**E**) Quantification of relative pSFK expression in TMA GBM samples; scale bar, 50 µm. (**F**) Quantification of relative expression of PDGFRA and PDGFRB in high pSFK tumour samples.

**Figure 5 cancers-13-00361-f005:**
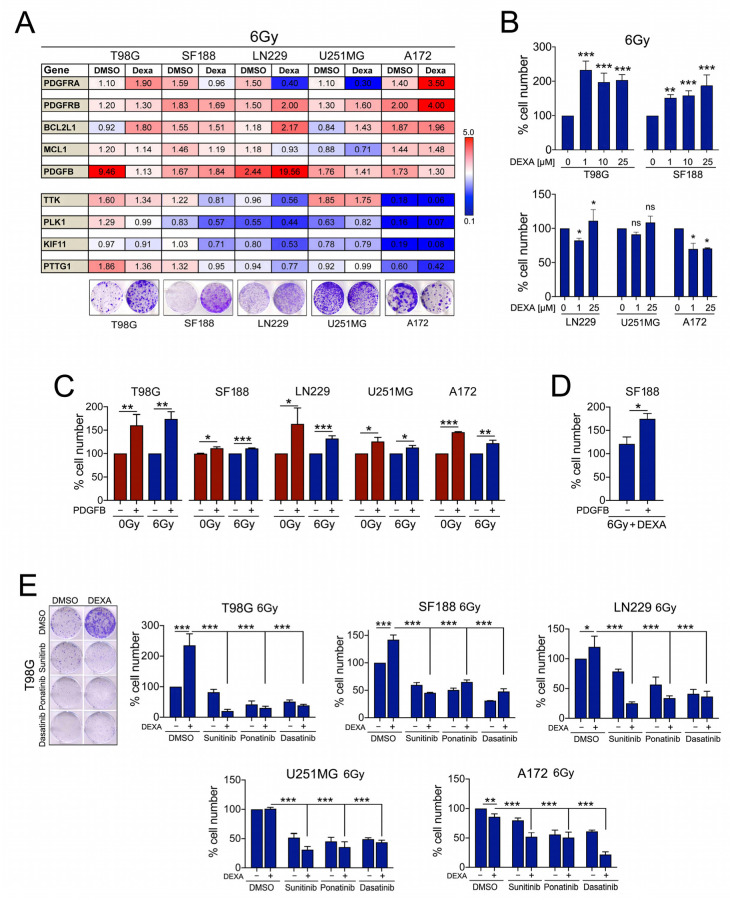
DEXA and PDGFB act as radio-protective factors and tyrosine kinase inhibitors (TKIs) inhibit this protection. (**A**) qRT-PCR analysis for the indicated genes in different GBM cell lines treated with 25 µM DEXA for 18 h before radiation with 6 Gy. RNA expression was analysed 24 h later. Data are represented as mean fold change of triplicates treated relative to non-radiated DMSO control cells (=1). A colony formation assay is shown as example for the radio-protective effect of DEXA in radiated cells. (**B**) Quantification of cell numbers in the indicated cell lines radiated with 6 Gy in the absence or presence of the indicated concentrations of DEXA as described above. Data represent the mean ± SEM (*n* ≥ 3). (**C**) Quantification of cell numbers in the indicated cell lines either non-radiated or radiated with 6 Gy in the absence or presence of recombinant PDGFBB. Data represent the mean ± SEM (*n* ≥ 3). (**D**) Quantification of cell numbers of SF188 cells radiated with 6 Gy and treated with DEXA in the absence or presence of recombinant PDGFBB. Data represent the mean ± SEM (*n* ≥ 3). (**E**) Quantification of cell numbers in the indicated cell lines radiated with 6 Gy in the absence or presence of DEXA and the indicated inhibitors. Data represent mean ± SEM (*n* ≥ 3). * *p* < 0.05, ** *p* < 0.01, *** *p* < 0.001.

**Figure 6 cancers-13-00361-f006:**
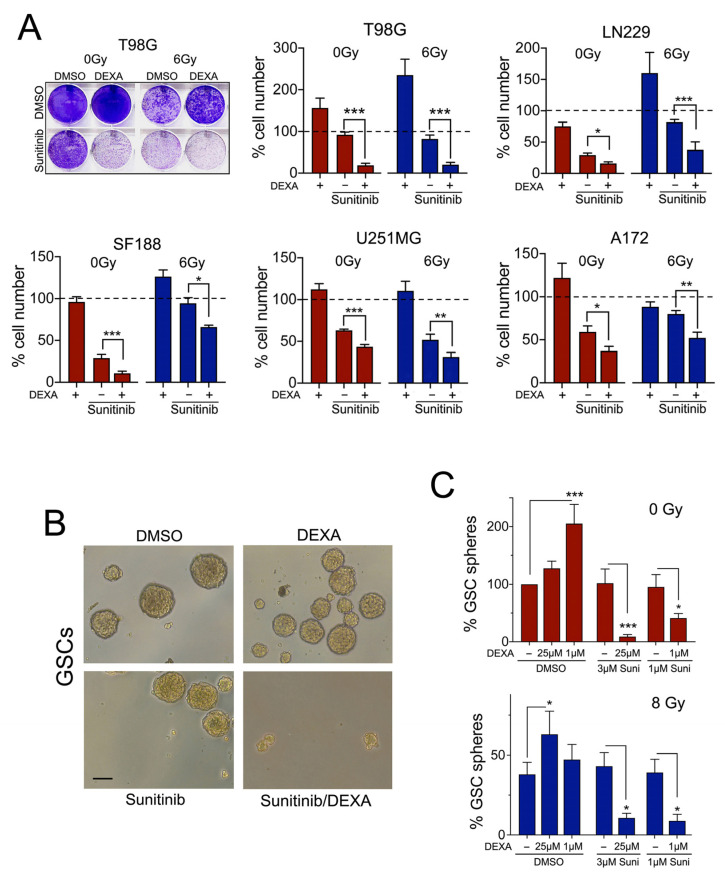
Dexamethasone sensitizes glioblastoma cells to sunitinib. (**A**) Quantification of cell numbers in T98G cells either non-radiated or radiated with 6 Gy in the absence or presence of DEXA and sunitinib. Data represent mean ± SEM (*n* ≥ 3). (**B**) Images depicting neurospheres formed by GSCs under the indicated conditions. (**C**) Quantification of sphere formation by GSCs cultured in stem cell medium under non-adherent conditions either non-radiated or radiated with 8 Gy in the absence or presence of DEXA or sunitinib (Sun). Data represent mean ± SEM (*n* ≥ 3). * *p* < 0.05, ** *p* < 0.01, *** *p* < 0.001.

**Figure 7 cancers-13-00361-f007:**
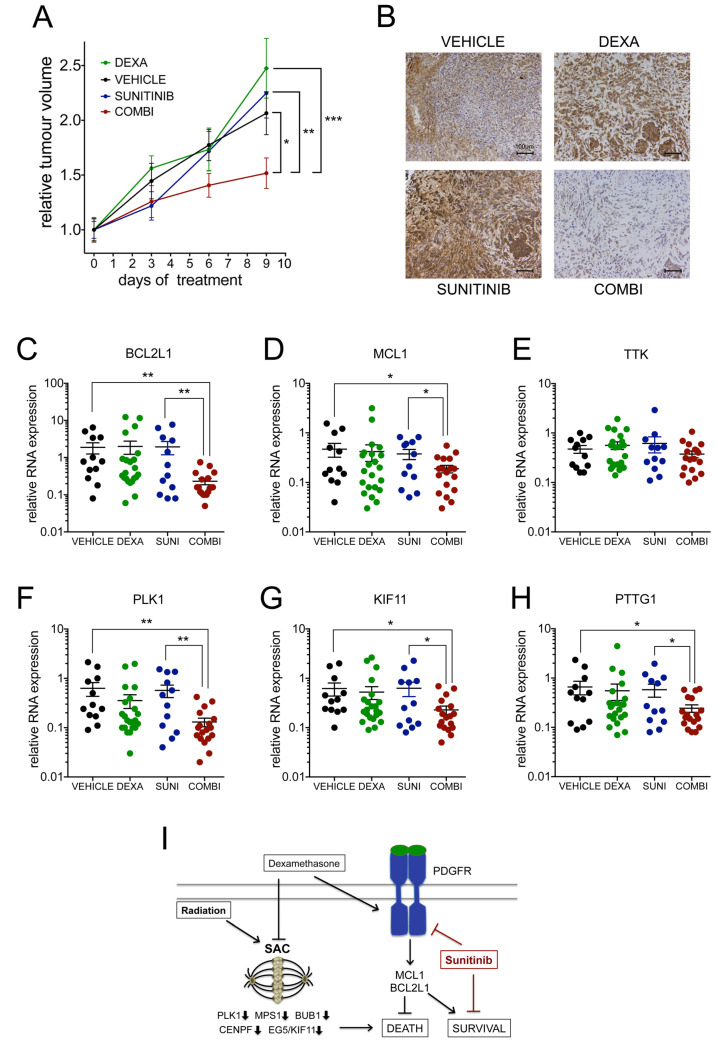
Dexamethasone sensitizes to sunitinib in vivo. (**A**) Mean tumour volumes ± SEM of mice (*n* ≥ 6 mice/group) treated as indicated: sunitinib (40 mg/kg/qd), DEXA (0.3mg/kg/qd). (**B**) IHC for phospho-SRC/SFK in the indicated tumours; scale bar 100 µm. (**C**–**H**) qRT-PCR analysis for the indicated genes of tumours from mice treated as described in (**A**). Data are from two experimental repeats of *n* = 6–11 tumours per group represented as scatter dot blot; the mean ± SEM is indicated. (**I**) Model of DEXA induced signalling that creates vulnerability for sunitinib. * *p* < 0.05, ** *p* < 0.01, *** *p* < 0.001.

## Data Availability

Data is contained within the article or Appendix A.

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
