# Peer review of "Identification of a Dexamethasone Mediated Radioprotection Mechanism Reveals New Therapeutic Vulnerabilities in Glioblastoma"

_cancers, 2021, doi:10.3390/cancers13020361_

Round 1

Reviewer 1 Report

The authors have adequately addressed the points listed in my previous review.

This is a fine study.

Author Response

We thank the reviewer for their comments

Reviewer 2 Report

The authors have satisfied my comments.

Author Response

We thank the reviewer for their comment

Reviewer 3 Report

Thanks to the authors to provide more experiments and address all the question.

Would the authors explain the inclusion of MC10a in the panel of cells assessed with DEXA?

Author Response

We apologise if we did not make this clear enough. We included MCF-10A cells, because one of the other reviewers requested their use in order to assess the impact of DEXA on the proliferation of non-cancerous cells.

As such, we had mentioned in the text (lines 178/179) that DEXA suppresses the proliferation of non-transformed MCF-10A cells.

This manuscript is a resubmission of an earlier submission. The following is a list of the peer review reports and author responses from that submission.

Round 1

Reviewer 1 Report

Aldaz and colleagues address an important and highly relevant topic: DEXA is frequently used to manage inflammation and control increased intracranial pressure of GBM patients. However, several clinical studies suggest that administration of DEXA restricts effective radiotherapy and correlates with poorer survival. In this manuscript the authors propose that DEXA directly protects GBM cells from radiotherapy possibly through transcriptional rewiring that culminates in the downregulation of G2/M genes and upregulation of PDGFR signalling, BCL2L1 and MCL1. The authors further claim that this DEXA-induced transcriptional profile probably underlies SAC override and enables GBM cells to cope with DNA damage and aneuploidy. Aldaz and colleagues show that this protective effect of DEXA can be prevented by impairing tyrosine kinase signalling. In vivo evidence indicates that sunitinib synergises with DEXA in reducing GBM tumour growth in mice.

Overall this is a good and comprehensive study. Apart from minor points, the manuscript is well written and clear. Most of the conclusions are supported by the data. The findings are relevant and will most likely represent important knowledge to the community working on GBM. I recommend the publication of this manuscript, but I would like to see the following points addressed/discussed by the authors:  

1-To strengthen and generalize the claim that DEXA leads to an increase in the number of mitotic cells, could the authors also provide the analyses of mitotic index and cell proliferation (FIG 2D and 2E) for other GBM cell lines? What is the impact of DEXA in the proliferation of non-cancerous cell lines (RPE1 or MCF-10A for instance)?    

2-To evaluate SAC proficiency, it would have been better to treat cells with vincristine in the presence or absence of DEXA (or reversine for control) and determine the mitotic index at t0h, t6h, t12h, t24h and 48h. The mitotic index should be determined by pSer10-H3 staining, which allows for a direct scoring of mitotic cells. In this way, the authors can directly examine weather DEXA prevents a vincristine-induced mitotic arrest, which would otherwise occur due to SAC activation.

Furthermore, to strengthen the conclusion that the SAC is weakened in DEXA-treated cells, the authors could measure and compare the levels of SAC proteins (ex. MAD2, MAD1 or BubR1) at unattached kinetochores in the presence and absence of DEXA treatment.

3-Line 207 the authors state: “The ability of DEXA to override the checkpoint most relevant for a cell to detect and eliminate DNA damaged cells would suggest that apart from driving basal proliferation it is capable to support the continued division of cells even if they have encountered radiation-induced DNA damage.” I find this sentence very confusing and difficult to follow. Could the authors please clarify.

4-Line 277 the authors state “We next wished to assess whether activation of the pathway we have identified to be relevant for DEXA mediated pro-proliferative activities could be detected in human glioblastoma.” Please, consider instead something like “We next assessed whether activation of the pathway we have identified was relevant for DEXA mediated pro-proliferative activities of human glioblastoma”

5- The authors conclude “We have identified a novel dexamethasone-induced mechanism that can directly protect GBM cells from radiotherapy and thus may contribute to the adverse effects observed in the clinic. Strikingly this mechanism also sensitises GBM cells to tyrosine kinase inhibitors, thus encouraging the revision of the use of these inhibitors for the treatment of GBM, potentially in an adjuvant setting.” I do agree that the synergy between DEXA and sunitinib might represent a promising therapeutic strategy. However, and in order to fuel the translation of this idea into a clinical setting, this study would gain considerably impact if the authors could also provide data on the effect of DEXA+sunitinib on the overall survival of mice with GBM tumours and more specifically of mice subjected to radiotherapy.    

6- I was surprise to see that in Fig 7C,D, DEXA treatment fail to cause the upregulation of BCL2L1 and MCL1. Could the authors discuss this result and further discuss why in some cell lines DEXA also fails to cause an increase in the expression of these genes (Figure 3G)?

Line 101: break-through > breakthrough

Line 106: radio-therapy > radiotherapy

Reviewer 2 Report

This is a well-performed study of the interactions of DEX with IR in glioblastoma, The authors show that DEX promotes radiation resistance, and upregulates protein kinase signaling pathways, and that these effects can be blocked and work well with sunitinib.  There is little that can be done to improve this work further.

Comments.

 The reasons for the choice of DEX dosing are not clear. In vivo, does these dose approximate that used in humans, and in vitro are there dose dependent effects?

Reviewer 3 Report

Aldaz et al in this work have reported in both in vitro and in vivo study the effect of DEXA in different GBM cells line showing how the combination of TKi and DEXA may efficiently kill GBM cells.

Overall the work has a clear aim based on good experimental design. Although the data are clearly reported I didn t find in supplementary section any document with RNAseq data from T98G used for the whole investigation. Did the authors perform RNA seq data on T98G (treated) vs (untreated)? would provide the list of the genes?

IHC could be reported better, the pictures are blurred and to small to look at the stainig. 

Would the authors summarize in the picture(cartoon) the molecular mechanisms proposed in this study ? Would be helpful for the reader to reach the point.

Since the authors have shown extensively the role of PDGFR and DEXA on GBM cells line. How they will explain the concurrent role of other glial cells which take part in the inflammatory process?

Do the authors have thought about the potential role of Microglia/macrophage and astrocytes in these processes?  The animal model does not reflect reality due to the place where cells have been injected however this experiment ould be clarified by using in vitro model at least to study the interaction with a stromal counterpart which play important role in GBM inflammatory process

However, the manuscript requires a profound editing along with grammar and spell check. 
